# Metagenomic and Untargeted Metabolomic Analysis of the Effect of *Sporisorium reilianum* Polysaccharide on Improving Obesity

**DOI:** 10.3390/foods12081578

**Published:** 2023-04-07

**Authors:** Yunlong Guo, Meihong Liu, Xin Liu, Mingzhu Zheng, Xiuying Xu, Xiaokang Liu, Jiyu Gong, Huimin Liu, Jingsheng Liu

**Affiliations:** 1National Engineering Research Center for Wheat and Corn Deep Processing, College of Food Science and Engineering, Jilin Agricultural University, Changchun 130118, China; guoyl02@ccucm.edu.cn (Y.G.); liuhuimin@jlau.edu.cn (H.L.); 2Jilin Ginseng Academy, Changchun University of Chinese Medicine, Changchun 130117, China; 3College of Pharmacy, Changchun University of Chinese Medicine, Changchun 130117, China

**Keywords:** *Sporisorium reilianum* polysaccharides, obesity, metabolomics, gut microbiota

## Abstract

Gut microbiota plays an important role in the pathophysiology of obesity. Fungal polysaccharide can improve obesity, but the potential mechanism needs further study. This experiment studied the potential mechanism of polysaccharides from *Sporisorium reilianum* (SRP) to improve obesity in male Sprague Dawley (SD) rats fed with a high-fat diet (HFD) using metagenomics and untargeted metabolomics. After 8 weeks of SRP (100, 200, and 400 mg/kg/day) intervention, we analyzed the related index of obesity, gut microbiota, and untargeted metabolomics of rats. The obesity and serum lipid levels of rats treated with SRP were reduced, and lipid accumulation in the liver and adipocyte hypertrophy was improved, especially in rats treated with a high dose of SRP. SRP improved the composition and function of gut microbiota in rats fed with a high-fat diet, and decreased the ratio of Firmicutes to Bacteroides at the phylum level. At the genus level, the abundance of Lactobacillus increased and that of Bacteroides decreased. At the species level, the abundance of *Lactobacillus crispatus*, *Lactobacillus helveticus*, and *Lactobacillus acidophilus* increased, while the abundance of *Lactobacillus reuteri* and *Staphylococcus xylosus* decreased. The function of gut microbiota mainly regulated lipid metabolism and amino acid metabolism. The untargeted metabolomics indicated that 36 metabolites were related to the anti-obesity effect of SRP. Furthermore, linoleic acid metabolism, phenylalanine, tyrosine, and tryptophan biosynthesis, and the phenylalanine metabolism pathway played a role in improving obesity in those treated with SRP. The study results suggest that SRP significantly alleviated obesity via gut-microbiota-related metabolic pathways, and SRP could be used for the prevention and treatment of obesity.

## 1. Introduction

The increasingly serious obesity epidemic is becoming a major public health problem, and the related costs pose a considerable burden to society [1]. The prevalence of obesity nearly tripled between 1976 and 2016. The World Health Organization predicts that by 2030, the number of overweight and obese people (including children and adults) in the world will reach 1.35 billion and 573 million, respectively [2]. The most common complications of severe obesity include hypertension [3], dyslipidemia [4,5], type 2 diabetes [6], cardiovascular diseases [7,8], and various types of cancer [9,10]. Edible fungi are one of the most widely distributed food resources in nature, with rich medicinal and nutritional values, and their bioactive components are increasingly valued by scientific researchers. Polysaccharides are one of their important active ingredients, and the latest studies have shown that they have good preventive, health care, and therapeutic effects, such as immunomodulation [11,12,13,14], anti-tumor [15,16,17,18], antioxidant [19,20,21], and hypoglycemic [22,23,24]. In addition, polysaccharides have a positive effect on the prevention of chronic metabolic diseases such as obesity, type 2 diabetes, and dyslipidemia [25]. The mechanism of polysaccharides to improve obesity mainly aims to inhibit fat absorption [26,27,28,29], improve the gut microbiota structure [30,31,32,33,34], and regulate bile acids [35,36,37].

*Sporisorium reilianum* was an edible fungus and folk medicine for many years which contains vitamins, proteins, and polysaccharides, and was rich in selenium, calcium, potassium, iron, magnesium, zinc, manganese, copper, and other minerals needed by the human body. It is mainly distributed in northeast China, north China, Gansu, Xinjiang, Shandong, Anhui, Henan, Hubei, Taiwan, Sichuan, Yunnan, and other provinces, municipalities, and autonomous regions [38]. Polysaccharides from *Sporisorium reilianum* have been shown to exert anti-colitis and anti-tumor effects, but studies on improving obesity are still scarce [39]. Considering the important role of gut microbiota in improving the glucose and lipid metabolism of obese hosts, we speculate that the changes in gut microbiota under the intervention of SRP may alleviate the obesity-related phenotype of rats fed with a high-fat diet. Accordingly, the main purpose of this study was to verify the interaction between fecal metabolites and gut microbiota through metagenomics and the untargeted metabolomics of feces, and then to clarify the potential mechanism of SRP to improve obesity induced by a high-fat diet from the perspective of “gut microbiota metabolism”. Meanwhile, the experimental results may provide a new perspective for functional food development and clinical research.

## 2. Materials and Methods

### 2.1. Preparation of Polysaccharides from S. reilianum

The powder of *Sporisorium reilianum* was collected from Jinmanwu Agricultural Science and Technology Development Co., Ltd. (Liaoyuan, China). The dried powder of *Sporisorium reilianum* was refluxed with 95% ethanol twice at 70 °C for 2 h to remove liposoluble compounds and impurities. After extraction twice, the combined extracts were concentrated, and were then subjected to deproteinization and decolorization according to the previous methods. Then, the resulting solution was precipitated (12 h, 4 °C) by adding dehydrated ethanol to a final concentration of 80% (*v*/*v*). Subsequently, the precipitate was obtained via centrifugation at 4000 rpm for 10 min. The precipitate was washed with anhydrous ethanol, acetone, and ether, and then dried to obtain SRP. Infrared spectrum showed that SRP had a general characteristic peak of polysaccharide at 4000~500 cm^−1^, and an O-H absorption vibration peak at 3418 cm^−1^, indicating that it contained polysaccharide. The C-H absorption vibration peak was at 2920 cm^−1^; the C=O absorption vibration peak was at 1633 cm^−1^; the C-N absorption vibration peak was at 1408 cm^−1^; 1047 cm^−1^ was the R-O absorption vibration peak; and 890 cm^−1^ was the characteristic absorption peak of C-H angular vibration. The above results show that SRP has the characteristic absorption peak of polysaccharide and has a β-pyranose structure (Appendix A).

### 2.2. Experimental Animals and Study Design

Male (six-week-old) SD rats were purchased from the Changchun Yisi Experimental Animal Technology Co., Ltd. (Changchun, China) under the animal experiment permission number SCXK(JI)-2020-0002. All experimental rats were kept in a specific-pathogen-free (SPF) animal laboratory where ambient environmental conditions (12 h light/dark cycle, temperature 25 ± 1 °C, relative humidity 50 ± 10%) were maintained, and the rats had free access to diet and water every day until the end of the study. Fifty rats were randomly divided into 5 groups (n = 10 per group) after 1 week of adaptation, including a normal control group, a model group, and three experimental groups. The normal control group (NC) was fed a normal diet and given normal saline every day. The other groups were fed with high-fat feed and given normal saline every day. After 8 weeks, 40 rats in the high-fat group were divided into four groups on average, namely the high-fat diet group and three experimental groups (HFD-SRPL, HFD-SRPM, and HFD-SRPH) which were given a high-fat diet and 100, 200, and 400 mg/kg/d SRP, respectively. All groups were given corresponding treatment, with continuous intervention for 8 weeks. The weight of the rats was measured once a week. After the experiment, the rats fasted for 12 h and were euthanized. The blood collected via cardiac puncture was centrifuged at 3500 rpm/min at 4 °C for 10 min to obtain the serum. Fecal samples were collected in sterile centrifugal tubes and were immediately frozen in liquid nitrogen and then stored at −80 °C for later analysis. The liver and white adipose tissues (perirenal and epididymal) were fixed in 4% paraformaldehyde for histopathological analysis. All animal experiments were approved by the laboratory animal ethics committee of Changchun University of Chinese Medicine (approval no. 2021553) and performed in accordance with the regulations and guidance of this committee.

### 2.3. Serum Lipids and Liver Enzyme Function Analysis

The levels of triacylglycerols (TGs), total cholesterol (TC), high-density lipoprotein cholesterol (HDL-c), low-density lipoprotein cholesterol (LDL-C), aspartate aminotransferase (AST), and alanine aminotransferase (ALT) in serum were measured using corresponding assay kits purchased from Shanghai Beyotime Biotechnology Co., Ltd. (Shanghai, China). 

### 2.4. Histological Analysis

The liver and white adipose tissue was fixed in 10% neutral formaldehyde for more than 24 h, and then the liver tissue and white adipose tissue in a good fixed state were trimmed in the fume hood. Histopathological sections and hematoxylin-eosin staining (HE staining) were prepared according to standard operating procedures.

### 2.5. DNA Extraction, Sequencing, and Analysis of Fecal Sample

About 150 mg of rat feces was extracted using the E.Z.N.A.^®^ Stool DNA Kit (Omega Bio-tek, Norcross, GA, USA) for bacterial DNA extraction according to the instructions of the kit manufacturer. The purity and concentration of extracted DNA were determined via NanoDrop2000 and TBS-380, respectively. The quality of DNA extraction was checked on 1% agarose gel. DNA extracted from fecal samples was fragmented to an average size of about 400 bp using Covaris M220 (Gene Company Limited, Hong Kong, China) for paired-end library construction. The paired-end library was constructed using NEXTFLEX Rapid DNA-Seq (Bioo Scientific, Austin, TX, USA). An adapter containing hybridization sites of all sequencing primers was attached to the blunt end of the fragment. Paired-end sequencing was performed using the Illumina HiSeq4000 platform (Illumina Inc., San Diego, CA, USA) at Majorbio Bio-Pharm Technology Co., Ltd. (Shanghai, China). The data were analyzed using the online platform Majorbio Cloud Platform (https://www.majorbio.com/, accessed on 5 November 2021). Sequence data associated with this project were deposited in the NCBI Short Read Archive database (accession number: PRJNA946687). The paired-end Illumina reads were trimmed of adaptors, and low-quality reads were removed using fastp (https://github.com/OpenGene/fastp, version 0.20.0, accessed on 6 November 2022) [40]. Reads were aligned to the rat genome via BWA (https://bio-bwa.sourceforge.net/, version 0.7.9a, accessed on 8 November 2022) and any hits associated with the reads and their mated reads were removed [41]. Metagenomic data were assembled using MEGAHIT (https://github.com/voutcn/megahit, version 1.1.2, accessed on 8 November 2022), which made use of succinct de Bruijn graphs [42]. Contigs with a length of 300 bp or above were selected as the final assembling results, and then the contigs were used for further gene prediction and annotation. Open reading frames from each assembled contig were predicted using MetaGene (http://metagene.cb.k.u-tokyo.ac.jp/, accessed on 10 November 2022) [43]. The predicted open reading frames with lengths of 100 bp or above were retrieved and translated into amino acid sequences using the NCBI translation table (https://www.ncbi.nlm.nih.gov/Taxonomy/taxonomyhome.html/index.cgi?chapter=tgencodes, accessed on 11 November 2022) and a non-redundant gene catalog was constructed using CD-HIT (http://www.bioinformatics.org/cd-hit/, version 4.6.1, accessed on 12 November 2022) with 90% sequence identity and 90% coverage. Reads after quality control were mapped to the non-redundant gene catalog with 95% identity using SOAPaligner (http://soap.genomics.org.cn/, version 2.21, accessed on 13 November 2022), and the gene abundance in each sample was evaluated [44,45]. Representative sequences of the non-redundant gene catalog were aligned to the NCBI NR database with an e-value cutoff of 1 × 10^−5^ using Diamond (https://github.com/bbuchfink/diamond, version 0.8.35, accessed on 13 November 2022) for taxonomic annotations. The clustering of orthologous groups of protein annotation for the representative sequences was performed using Diamond (https://github.com/bbuchfink/diamond, version 0.8.35, accessed on 13 November 2022) against the eggNOG database with an e-value cutoff of 1 × 10^−5^. The KEGG annotation was conducted using Diamond https://github.com/bbuchfink/diamond, version 0.8.35, accessed on 15 November 2022) and was compared against the Kyoto Encyclopedia of Genes and Genomes database (https://www.genome.jp/kegg/, accessed on 16 November 2022) with an e-value cutoff of 1 × 10^−5^ [46].

### 2.6. Sample Preparation and LC-HRMS Analysis

Fecal samples were thawed at 4 °C prior to analysis. A total of 400 μL of the mixed solution (methanol, acetonitrile, and water, 2:2:1, *v*/*v*) was added to a 100 mg sample and vortexed thoroughly. The samples were centrifuged at 12,000 rpm/min at 4 °C and this was repeated twice. The obtained supernatants of samples were passed through a 0.22 μm filter membrane to the sample bottle for LC-HRMS analysis. All samples were mixed in equal volumes to form a quality control sample (QC) to ensure the reliability of the analysis results. Chromatographic separation was performed using an Ultimate 3000 ultra-high-performance liquid chromatography system (Thermo, San Jose, CA, USA) coupled with a Supelco C_18_ column (3.0 × 50 mm, 2.7 μm; Sigma-Aldrich, St. Louis, MO, USA). The column temperature was maintained at 35 °C and 5 μL of each sample solution was injected. The temperature of the sample chamber was 4 °C. The mobile phases consisted of phase A (water with 0.1% formic acid) and phase B (100% acetonitrile) with a flowing rate of 0.4 mL/min. The gradient elution conditions were set as follows: 0–3 min, A 95–75%; 3–10 min, A 75–40%; 10–15 min, A 60–5%; 15–18 min, A 5–95%, 18–20 min, A 95%. High-resolution mass spectrometry detection was carried out using an Orbitrap-MS (Thermo, San Jose, CA, USA) equipped with an electrospray ionization source (ESI). The parameters of the ion source were set to 35 Arb for sheath gas flow, 10 Arb for aux gas flow, and 1 Arb for sweep gas flow. The S-Lens RF was 55%. The capillary voltage was set to 3.5 kV with a capillary temperature of 350 °C. Full MS data were acquired in the profile mode from m/z 100 to 1000 Da, using the resolution of 70,000, with an automatic gain control (AGC) target of 1 × 10^6^ and maximum injection time (IT) of 100 ms. The original data were preprocessed using Analysis Base File Converter and MS-DIAL software, including the retention time, peak intensity, and molecular weight of each sample. The normalized data were processed using unsupervised principal component analysis (PCA) with SIMCA-P (Version 13.0, Umetrics, Sweden) software, and then the data were clustered in a reduced dimension. Furthermore, the supervised partial least squares discriminant analysis (PLS-DA) and *t*-test were used to screen the significantly changed metabolites, and the *p* value was calculated according to the peak intensity. Differential metabolites were screened according to VIP > 1 and *p* < 0.05 standards, and compared with KEGG (https://www.genome.jp/kegg/, accessed on 7 May 2022) and HMDB (https://hmdb.ca/, accessed on 9 May 2022). Bioincloud.tech (https://www.bioincloud.tech/, accessed on 10 May 2022) and MetaboAnalyst 4.0 (https://www.metaboanalyst.ca/, accessed on 13 May 2022) were used for the statistical analysis and pathway enrichment analysis of differential metabolites.

### 2.7. Statistical Analysis

All data were expressed as mean ± SEM. Student’s *t*-test via the GraphPad Prism 8.0 software was used to analyze the significance of differences between the two groups. One-way ANOVA and Tukey multiple comparison analysis were used to analyze the significance between multiple groups.

## 3. Results

### 3.1. The Effect of SRP on the Weight of Obese Rats

The experimental results (Figure 1a) showed that the weight of experimental rats in the HFD group was significantly higher than in the NC group after 8 weeks of high-fat diet intervention. After the experiment, the weight gain of rats in the NC, HFD, HFD-SRPL, HFD-SRPM, and HFD-SRPH groups was 406.6 g/rat, 612.6 g/rat, 527.8 g/rat, 485.8 g/rat, and 443.1 g/rat, respectively. Compared with the HFD group, the weight gain of experimental rats decreased after they were fed with high, medium, and low doses of SRP. Compared with the HFD group, the weight gain of the rats in the NC, HFD-SRPL, HFD-SRPM, and HFD-SRPH groups decreased by 50.66%, 16.07%, 26.10%, and 38.25%, respectively. The results showed that SRP could improve the weight gain of obese rats with obesity induced by a high-fat diet.

### 3.2. Effect of SRP on Serum Lipid Levels and Liver Enzyme Functions

The serum lipid levels and liver enzyme functions of experimental rats in each group are shown in Figure 1b–g. The contents of TG (2.24 ± 0.15) and TC (3.89 ± 0.26) in the HFD group were the highest, which were 1.44 times and 2.35 times higher than those in the NC group, respectively, and therefore significantly higher than those in the NC group. Compared with the HFD group, the TG content in the HFD-SRPL, HFD-SRPM, and HFD-SRPH groups decreased by 23.05%, 28.53%, and 29.68%, respectively, with significant differences. In addition, the TC content in the HFD-SRPL, HFD-SRPM, and HFD-SRPH groups decreased by 31.81%, 47.73%, and 54.02%, respectively, with significant differences. The results of LDL-c showed that the content of LDL-C in the HFD group was the highest (0.77 ± 0.06), which was 2.03 times that in the NC group. Compared with the HFD group, the contents of LDL-c in the HFD-SRPL, HFD-SRPM, and HFD-SRPH groups decreased by 0.56%, 19.34%, and 35.45%, respectively. Except for the HFD-SRPL group, there were significant differences among the other groups. The HDL-c results showed that the HDL-c contents in the HFD group (0.44 ± 0.05) was the lowest, and that that in the NC group was 1.73 times higher. Compared with the HFD group, the HDL-c content in the HFD-SRPL, HFD-SRPM, and HFD-SRPH groups increased by 18.09%, 28.78%, and 35.10%, respectively, with significant differences. The contents of ALT (99.70 ± 8.63) and AST (245.60 ± 16.51) in the HFD group were the highest, which were 2.25 times and 1.75 times higher than those in the NC group, respectively. Compared with the HFD group, the ALT content in the HFD-SRPL, HFD-SRPM, and HFD-SRPH groups decreased by 38.31%, 48.85%, and 52.19%, respectively, with significant differences. Compared with the HFD group, the AST content in the HFD-SRPL, HFD-SRPM, and HFD-SRPH groups decreased by 21.42%, 32.25%, and 45.85%, respectively, with significant differences. The above results indicate that SRP can improve the serum lipid levels and liver function of obese rats with obesity induced by a high-fat diet.

### 3.3. Pathological Analysis of Rat Liver and White Adipose Tissue Sections

The liver sections from rats in the NC group showed normal hepatocyte morphology without obvious degeneration (Figure 2a). In contrast, the sections from the HFD group displayed obvious symptoms of lipid degeneration, such as fat droplets. These results indicate that SRP could improve lipid deposition and the fatty lesions of hepatocytes induced by a high-fat diet, particularly in the HFD-SRPH group.

The white adipose tissue sections from rats in the NC group showed that the fat cells had a uniform size and a regular polygonal structure that was closely arranged and clearly visible (Figure 2b). However, in the HFD group, the size of adipocytes increased obviously, the cells tended to fuse with each other, and the shape of the cells became irregular. These results indicate that SRP could improve the degeneration of white tissue cells in obese rats with obesity induced by a high-fat diet.

### 3.4. Quality Control of Fecal Metagenomics

The results of sequencing the raw data are shown in Appendix A. The average value of the raw reads is 51,539,591, and the average value of the raw base (bp) is 7,782,478,241. The results of the sequencing data are shown in Appendix A. The average percent in raw reads is 97.80%, and the average percent in raw bases is 97.64%. The results of the gene sequencing prediction data are shown in Appendix A, with an average ORF of 563,992 and an average total length (bp) of 298,928,983. These results show that the data quality is reliable and that the library is well constructed, which is in accordance with the expected effect. 

### 3.5. Effect of SRP on Gut Microbiota

The principal coordinate analysis (PCoA) results based on the Bray–Curtis of the three groups of fecal samples are shown in Figure 3a–c, and the distances between the samples of the NC and the HFD groups at the phylum, genus, and species levels were far. The results showed that a high-fat diet changed the composition of gut microbiota. After SRP intervention, there was some separation between the HFD-SRPH and HFD groups, especially at the species level with a tendency to approach the NC group, indicating that SRP could improve the composition of gut microbiota in rats on a high-fat diet.

At the phylum level, compared with the NC group, the abundance of Firmicutes in the HFD group increased and that of Bacterioidetes decreased (Figure 4a). After the intervention of SRP, the abundance of Firmicutes in the HFD-SRPH group decreased and that of Bacterioidetes increased. At the same time, the ratio (F/B) of Firmicutes/Bacterioidetes in the NC group, HFD group, and HFD-SRPH group was 2.16, 7.65, and 4.28, respectively. High-dose SRP decreased the ratio of F/B in rats fed with a high-fat diet. As seen via the analysis of the above results, SRP can improve the intestinal microbial structure at the phylum level.

At the genus level, compared with the NC group, the abundance of Lactobacillus in the HFD group decreased and that of Bacteroides increased (Figure 4b). After the intervention of *Sporisorium reilianum* polysaccharide, the abundance of Lactobacillus in the HFD-SRPH group increased and that of Bacteroides decreased.

According to read abundance data, the species with significant differences among groups and the abundances of the top 15 species were obtained (Figure 4c). In descending order of abundance, they were *Lactobacillus crispatus*, *Lactobacillus reuteri*, *Lactobacillus* sp. *ASF360*, *Alistipes* sp. *Z76*, *Staphylococcus xylosus*, *Lactobacillus helveticus*, *Lactobacillus amylovorus*, *Lactobacillus acidophilus*, *Bacteroides* sp. *CAG:927*, *Barnesiella* sp. *WM24*, *Clostridium* sp. *CAG:678*, *Clostridium* sp. *CAG:343*, *Ruthenibacterium lactatiformans*, *Prevotella* sp. *CAG:485*, and *Prevotella* sp. *CAG:873*. Eleven species of them were reversed, and these were *Lactobacillus crispatus*, *Lactobacillus reuteri*, *Staphylococcus xylosus*, *Lactobacillus helveticus*, *Lactobacillus amylovorus*, *Lactobacillus acidophilus*, *Bacteroides* sp. *CAG:927*, *Barnesiella* sp. *WM24*, *Clostridium* sp. *CAG:678*, *Prevotella* sp. *CAG:485*, and *Prevotella* sp. *CAG:873*.

As shown in Figure 4d, the top 15 related pathways with significant differences between groups were obtained, which were arranged in descending order of abundance as follows: Cysteine and methionine metabolism, Pentose phosphate pathway, Glycerolipid metabolism, Nicotinate and nicotinamide metabolism, Riboflavin metabolism, Inositol phosphate metabolism, Ascorbate and aldarate metabolism, protein digestion and absorption, Retinol metabolism, the metabolism of xenobiotics via cytochromem, D-Arginine and D-ornithine metabolism, steroid hormone biosynthesis, alpha-Linolenic acid metabolism, Linoleic acid metabolism, and non-alcoholic fatty liver disease. Thirteen pathways of them were called back, and these were Cysteine and methionine metabolism, Glycerolipid metabolism, Nicotinate and nicotinamide metabolism, Riboflavin metabolism, Inositol phosphate metabolism, Ascorbate and aldarate metabolism, protein digestion and absorption, Retinol metabolism, the metabolism of xenobiotics via cytochrome P450, steroid hormone biosynthesis, alpha-Linolenic acid metabolism, Linoleic acid metabolism, and non-alcoholic fatty liver disease.

### 3.6. Analysis of Metabolic Profile of Untargeted Metabolomics in Feces

SIMCA-P was used to analyze the data. The principal component analysis (PCA) three-dimensional scores plot is an unsupervised statistical analysis method that can visually process the different metabolites of three groups of samples (Figure 5a,e). Each dot in the figure represents a biological sample. The distribution of each dot in the figure indicates that there is an obvious separation trend among the fecal samples of the NC, HFD, and HFD-SRPH groups, and it also suggests that there are different degrees of differences among the three groups of fecal samples. The NC, HFD, and HFD-SRPH groups have an obvious separation in the positive and negative ion modes. Compared with the HFD group, the HFD-SRPH group is closer to the NC group. QC samples are gathered together in the PCA three-dimensional plot, which ensures the stability of the system in the processes of the experiment and data acquisition. In order to screen the different metabolites among the NC, HFD, and HFD-SRPH groups, a supervised partial least squares discriminant analysis (PLS-DA) model was established (Figure 5b–d,g,h). The displacement test plot in positive and negative ion modes shows that the positive ion modes are R^2^Y = 0.987 and Q^2^ = 0.935, and the negative ion modes are R^2^Y = 0.989 and Q^2^ = 0.945. R2Y reflects the stability of the model, and the closer the value is to 1, the better the stability of the model. Q^2^ reflects the reliability of the model. The closer the numerical value is to 1, the better the reliability of the model.

The differential metabolites screened in the above multivariate statistical analysis (VIP > 1, *p* < 0.05) were identified via the HMDB database. A total of 36 differential metabolites were identified via positive and negative ion modes (Appendix A). Compared with the NC group, the levels of 11 different metabolites in HFD increased and 25 different metabolites decreased. After SRP intervention, compared with HFD, the levels of 19 different metabolites in the HFD-SRPH group increased and 17 different metabolites decreased; 24 different metabolites can be reversed.

The 36 identified differential metabolites were inputted into the pathway analysis column on the website of Metabo Analyst (https://www.metaboanalyst.ca/, accessed on 13 May 2022) to enrich and analyze the metabolic pathways (Figure 6a). According to the *p* value and impact value, the metabolic pathways related to obesity were screened (Appendix A). The results showed that the main metabolic pathways were linoleic acid metabolism, phenylalanine, tyrosine, and tryptophan biosynthesis, phenylalanine metabolism, arachidonic acid metabolism, sphingolipid metabolism, tryptophan metabolism, and inositol phosphate metabolism.

### 3.7. Correlation between Gut Microbial Species and Differential Metabolites

In order to study the relationship between fecal metabolites and gut microbiota in rats after the intervention of SRP, we analyzed Spearman’s correlation between 24 different metabolites and gut microbiota (species level), and the results are shown in Figure 6b and Appendix A. *Barnesiella* sp. *WM24*, *Bacteroides* sp. *CAG:927*, *Prevotella* sp. *CAG:485*, *Prevotella* sp. *CAG:873*, *Lactobacillus crispatus*, *Lactobacillus acidophilus*, and *Lactobacillus helveticus* were positively correlated with myo-Inositol and Oxalosuccinic acid and negatively correlated with Alanyltryptophan, Cyclic AMP, LysoPC(14:0/0:0), Lithocholic acid, Glycocholic acid, Docosahexaenoic acid, Stercobilinogen, Chenodeoxycholic acid, Dimorphecolic acid, LysoPC(17:0/0:0), and Deoxycholic acid. *Clostridium* sp. *CAG:678*, *Staphylococcus xylosus*, and *Lactobacillus reuteri* were positively correlated with myo-Inositol and Oxalosuccinic acid and negatively correlated with Alanyltryptophan, Cyclic AMP, LysoPC(14:0/0:0), Lithocholic acid, Glycocholic acid, Docosahexaenoic acid, Stercobilinogen, Chenodeoxycholic acid, Dimorphecolic acid, LysoPC(17:0/0:0), and Deoxycholic acid. These results suggest that certain gut microbiota that were shifted by SRP were correlated with metabolites in linoleic acid metabolism, phenylalanine, tyrosine, and tryptophan biosynthesis, and phenylalanine metabolism, indicating the important roles of these gut microbiota in SRP-associated beneficial effects.

## 4. Discussion

The imbalance between fat storage and consumption in the body is the main cause of obesity, which leads to a large amount of fat accumulation in the body [47]. Under normal circumstances, there is a positive correlation between obesity and weight. In this experimental study, SRP was seen to significantly reduce the weight gain of SD obese rats, with obesity induced by a high-fat diet. It can be inferred that SRP can significantly reduce the weight of SD obese rats with obesity induced by a high-fat diet.

Triglycerides are mainly involved in the energy metabolism process, and cholesterol also plays an important role in the synthesis of bile acids, steroid hormones, and cell membranes [48,49]. Obesity can be understood as a metabolic syndrome, which is usually accompanied by dyslipidemia in the body, which mainly induces symptoms such as hypercholesterolemia, hypertriglyceridemia, and low-density lipoprotein [50], while the existence of dyslipidemia leads to a high risk of hypertension, coronary heart disease, and cardiovascular and cerebrovascular diseases [51]. A long-term high-fat diet will obviously increase the contents of triglyceride, total cholesterol, and low-density lipoprotein cholesterol, showing the characteristics of hyperlipidemia. However, after the intervention of SRP, the contents of triglyceride, total cholesterol, and low-density lipoprotein cholesterol in the serum of rats decreased significantly. Therefore, SRP can effectively improve the dyslipidemia caused by a high-fat diet in rats, and has a positive preventive effect on cardiovascular and cerebrovascular diseases. 

The liver plays an important role in the process of fat metabolism in the body. A high-fat diet leads to the disorder of liver metabolism, which leads to an accumulation of fat in liver cells, which makes the liver slightly fatty and it can even develop into a non-alcoholic fatty liver [52]. When the liver is abnormal, the levels of alanine aminotransferase and aspartate aminotransferase change, which leads to an increase in blood content. Therefore, alanine aminotransferase and aspartate aminotransferase can be used to evaluate whether the liver function is abnormal [53]. A high-fat diet significantly increased the contents of alanine aminotransferase (ALT) and aspartate aminotransferase (AST) in the serum of rats, suggesting that the liver function of experimental rats was seriously damaged. The content of ALT and AST decreased significantly after the intervention of SRP. A histopathological section of liver showed that the liver lesions were reversed after the intervention of SRP, which further verified the effects of changes in ALT and AST on liver health, suggesting that SRP could effectively improve liver tissue damage and play a role in protecting the liver.

The number and volume of fat cells in the body have an important influence on the content of fat in the body. The number and volume of fat cells in a normal body are smaller than those in an obese body [54]. Through the research, it was found that, compared with the HFD group, after SRP intervention the number of adipocytes increased significantly and the size decreased significantly. The morphology of adipocytes is closer to that of the NC group with an increase in the intragastric dose of SRP. Therefore, it can be inferred that a decrease in lipid droplet accumulation in adipocytes is due to the direct or indirect inhibition of fatty acid synthesis by SRP.

Obesity induced by a high-fat diet is closely related to metabolic disorder, chronic low-grade inflammation, and structural changes in gut microbiota [55,56]. In the body, gut microbiota cannot only digest food, but also regulate the microbial ecosystem. The structural imbalance in gut microbiota is related to the occurrence of metabolic syndromes such as being overweight and obesity [57].

In this experiment, compared with the NC group, the composition and function of gut microbiota in the HFD group changed, but they recovered with the intervention of SRP. Bacteroidetes and Firmicutes played an important role in rat gut microbiota at the phylum levels, and they were the main gut microbiota involved in dietary fiber and polysaccharide metabolism [58]. The intervention of SRP can reduce the relative abundance of Firmicutes and increase the level of Bacteroides in obese rats with obesity induced by a high-fat diet, and then reduce the F/B value. In the study of obesity induced by a high-fat diet in rats, a reduction in the ratio of Firmicutes to Bacteroidetes can effectively control the development of obesity, and the ratio of Firmicutes to Bacteroides is significantly reduced compared with obese individuals [59,60]. The intervention of SRP can increase the relative abundance of Lactobacillus and reduce the level of Bacteroides in obese rats with obesity induced by a high-fat diet, and increase the content of Lactobacillus in the adult population struggling with weight loss [61,62,63]. The proportion of Bacteroides in obese people is relatively high, and it is positively correlated with body mass index (BMI) [63]. We enriched the top 15 strains in abundance. As a result, a high-fat diet increased the abundance of one strain and decreased the abundance of ten strains. SRP can reverse these changes. Although some lactobacilli are related to weight gain, most lactobacilli have anti-obesity effects [64]; *Lactobacillus reuteri* is a flora that can promote weight gain [65], and *Lactobacillus acidophilus* can reduce the level of triglycerides and the activities of aspartate aminotransferase and alanine aminotransferase, and significantly inhibit the development of obesity and lipid deposition in the liver of ICR mice fed with HFD [66]. Clostridium prevents weight gain by blocking the intestinal absorption of fat, and experiments have also confirmed that mice that make Clostridium the only living bacteria in their intestinal tract are thinner and less fat than mice that have no microbial flora at all [67]. A study showed that Lactobacillus amyloliquefaciens promoted the browning of white adipocytes via the PPAR-PGC-1α transcription complex, at least partly by increasing the lactic acid level, which led to the inhibition of diet-induced obesity [68]. Anti-obesity gut microbiota, such as Bifidobacterium, Lactobacillus, and Bacteroides, can induce weight loss in many ways, including reducing intestinal permeability, enhancing the integrity of intestinal mucosa by increasing the level of the tight junction protein, regulating the expression of key regulatory factors involved in fat production, reducing intestinal inflammation, reducing insulin resistance, and promoting the browning of white fat by reducing the levels of TNF-α and IL-1β. It should be clear that different species of the same genus may have different effects on obesity. Abnormal amino acid metabolism was related to irritable bowel syndrome, metabolic syndrome, obesity, infectious diseases, and neuropsychiatric disorders [69,70]. Glycerol phospholipid metabolism is closely related to obesity, which can regulate obesity induced by a high-fat diet and obesity-related complications, such as hyperlipidemia, cardiovascular disease, and a non-alcoholic fatty liver [71]. We found that a high-fat diet increased six enrichment pathways and decreased seven enrichment pathways, and SRP could reverse these changes. The metabolism of cysteine and methionine and the metabolism of glycerol and lipid were significantly enhanced by a high-fat diet, but reversed by the intervention of SRP, which indicates that the gut microbiota functions of SRP are mainly lipid metabolism and amino acid metabolism in obese rats with obesity induced by a high-fat diet.

In this experiment, LC-HRMS technology was used to analyze the fecal metabolic profiles of rats in NC, HFD, and HFD-SRPH groups, to screen differential metabolites, and to enrich related pathways. A total of 36 fecal differential metabolites related to obesity were identified, and among which 24 differential metabolites were reversed. Among them, a high-fat diet increased the relative contents of 8 differential metabolites and decreased the relative contents of 16 differential metabolites, and SRP could reverse these changes. The impact value represents the role of the selected metabolites in the pathway. The larger the value, the greater the role of the concerned differential metabolites in the metabolic pathway, which is at the key node. According to the impact value, the top three enriched metabolic pathways were selected. The results showed that obesity may be related to linoleic acid metabolism (impact value: 1.00), phenylalanine, tyrosine, and tryptophan biosynthesis (impact value: 0.50), phenylalanine metabolism (impact value: 0.36), arachidonic acid metabolism (impact value: 0.36), and tryptophan metabolism (impact value: 0.36). In order to further study this, we enriched 24 compounds that can be reversed, and the pathways obtained via enrichment were linoleic acid metabolism, phenylalanine, tyrosine, and tryptophan biosynthesis, and phenylalanine metabolism. 

Linoleic acid metabolism: Linoleic acid is a polyunsaturated fatty acid, and it is also an essential fatty acid for the body. At the same time, Linoleic acid is the precursor of arachnoid, but the arachnoid derived from Linoleic acid is usually pro-inflammatory [72]. The therapeutic effect of obesity involves the linoleic acid metabolic pathway, the α -linolenic acid metabolic pathway, the glycerophosphate metabolic pathway, the arachidonic acid metabolic pathway, and the pyrimidine metabolic pathway [73]. This experimental study shows that SRP can regulate linoleic acid metabolism to improve the obesity caused by a high-fat diet intake.

Biosynthesis of phenylalanine, tyrosine, and tryptophan: There is evidence that excessive branched-chain amino acids can cause insulin resistance and glucose metabolism disorder [74]. At the same time, aromatic amino acids, especially phenylalanine and tyrosine, in the plasma of obesity-related insulin resistance patients increased, together with branched-chain amino acids [75]. In addition to the changes in amino acids in plasma, some studies have confirmed that fecal metabolomics can analyze the metabolic damage caused by obesity, which is mainly manifested by the increase in fecal amino acids such as valine, alanine, phenylalanine, and tyrosine [76]. In this study, compared with the normal group, the level of phenylalanine in the feces of the high-fat diet group increased, and the level decreased after the intervention of SRP, indicating that SRP can reduce the weight of obese rats by regulating the biosynthesis of phenylalanine, tyrosine, and tryptophan. Phenylpropionic acid metabolism: SRP can regulate phenylalanine metabolism to reduce the content of phenylalanine in the feces of rats fed with a high-fat diet, thus reducing the weight of obese rats. In the above, SRP has a significant effect on gut microbiota and metabolites in rats fed with a high-fat diet, which can effectively increase the abundance of some beneficial bacteria or beneficial metabolites. However, there are still some limitations to be further studied in the regulation of host metabolism via SRP through gut microbiota. Sterile animal models or fecal transplantation may be needed to verify whether the changes in small molecular metabolites in the gut microbiota and feces of the host after SRP intervention are caused by the changes in gut microbiota. In the future, an in-depth study of these aspects may provide a new perspective for functional food development and clinical research.

## 5. Conclusions

In this study, we demonstrated that SRP can improve HFD-induced obesity and regulate gut microbiota and metabolism. Specifically, SRP significantly reduced body weight, lipid accumulation in the liver, and adipocyte hypertrophy of HFD rats, especially in high doses. In addition, SRP interventions improved serum lipid levels. Gut microbiota and metabolomic analysis showed that SRP regulated the community structure of the gut microbiota and changed the structure at the metabolite levels. Therefore, our study provides detailed insights into the role of SRP in improving obesity.

## Figures and Tables

**Figure 1 foods-12-01578-f001:**
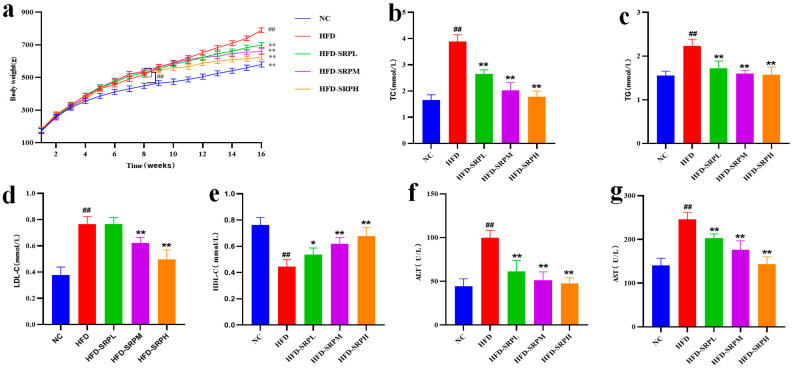
SRP improved HFD-induced obesity and metabolic disorders. (**a**) Body weight. (**b**) Serum total cholesterol. (**c**) Serum triacylglycerols. (**d**) Low-density lipoprotein cholesterol. (**e**) High-density lipoprotein cholesterol. (**f**) Alanine aminotransferase. (**g**) Aspartate aminotransferase. One-way ANOVA and Tukey multiple comparison analysis were used to analyze the significance between multiple groups compared with NC, ## *p* < 0.01, and compared with HFD, * *p* < 0.05, ** *p* < 0.01.

**Figure 2 foods-12-01578-f002:**
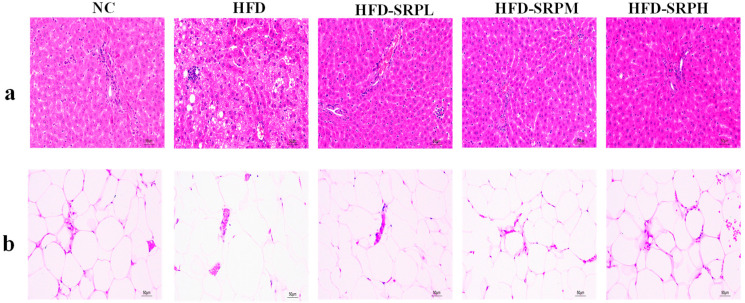
Pathological sections of liver and white adipose tissue were stained with HE (×200). (**a**) Liver tissue. (**b**) White adipose tissue.

**Figure 3 foods-12-01578-f003:**
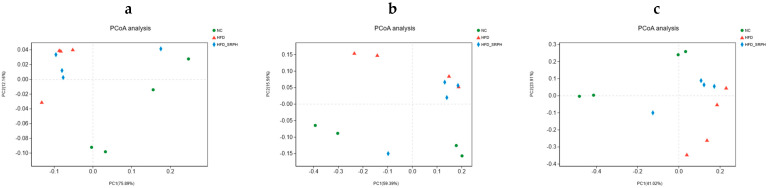
Fecal samples of NC, HFD, and HFD−SRPH groups were analyzed using PCoA. (**a**) PCoA of phylum. (**b**) PCoA of genus. (**c**) PCoA of species.

**Figure 4 foods-12-01578-f004:**
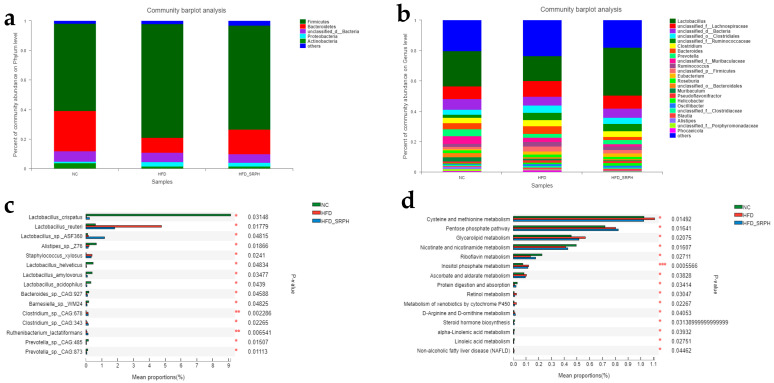
The fecal samples of NC, HFD, and HFD-SRPH groups were analyzed according to composition and function. (**a**) Relative abundance of taxonomic species at the levels of phylum. (**b**) Relative abundance of taxonomic species at the levels of genus. (**c**) Relative abundance of taxonomic species at the level of species. (**d**) Gut microbial function in HFD-fed mice. * *p* < 0.05, ** *p* < 0.01, *** *p* < 0.001.

**Figure 5 foods-12-01578-f005:**
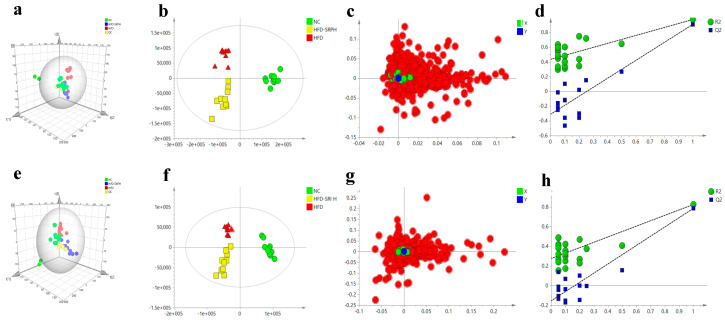
SRP changes fecal metabolome in HFD-fed rats. The LC/MS-based untargeted metabolic profiling in positive and negative modes was performed on fecal samples. (**a**) PCA of positive ion mode. (**b**) PLS-DA in positive ion mode. (**c**) Loading plot of positive ion mode. (**d**) Permutation plot of positive ion mode. (**e**) PCA of negative ion mode. (**f**) PLS-DA in negative ion mode. (**g**) Loading plot of negative ion mode. (**h**) Permutation plot of negative ion mode.

**Figure 6 foods-12-01578-f006:**
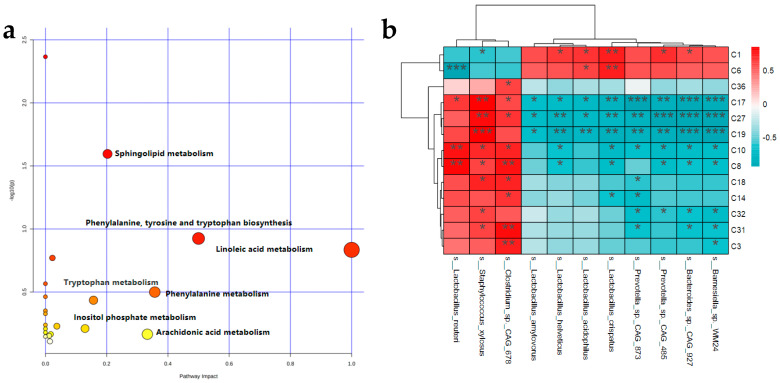
Metabolic pathways and correlation analysis of different metabolites. (**a**) Analysis of metabolic pathways of feces. (**b**) Spearman correlations between differential metabolites and differential bacterial species. Positive correlations indicated by red cubes and negative correlations indicated by blue cubes. * *p* < 0.05, ** *p* < 0.01, *** *p* < 0.001.

## Data Availability

Data are contained within the article or Appendix A.

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
