# Peer review of "Metagenomic and Untargeted Metabolomic Analysis of the Effect of Sporisorium reilianum Polysaccharide on Improving Obesity"

_foods, 2023, doi:10.3390/foods12081578_

Round 1

Reviewer 1 Report

The authors evaluate the effect of a fungus polysaccharide on obesity using rats model. For this purpose, they study serum biomarkers, gut microbiota community structure and untargeted metabolomics. The study design is appropiate, however I do have some comments:

*statistical methods are missing some pieces such as PCoA, correlations, etc. All statistical analysis, including those performed on metabolites, should be under the proper section.

*Pathway analysis is not described

*There is very little description on the DNA analysis. I assume is 16S by the lenght of the reads but is not specified. Also, how did you assigned taxonomies, did you grouped them in ASVs, OTUs? Which primers did you used? All the information is needed should the experiment be replicated

*Reported results regarding PCoA is kind of misleading. First, there are too few points when there should be 50 (50 rats). Second, there is no closeness between high fat diet + polysaccharide and normal diet, they are far apart and much closer to just high fat diet. Regardless, you are making an interpretation that is totally subjective, and it is hard to tell whether there is any  tendency as you claim. PCoA is just an exploratory method and unless you perform any additional statistical test that provides p values, you cannot make such an interpretation. 

You say: "The above results showed that obesity caused changes in the gut microbiota of rat feces, and showed that SRP could effectively regulate the gut microbiota of rat feces to achieve weight loss and lipid reduction". That is not supported at all by PCoA. You are just showing that introducing a polysaccharided midly changes gut microbiota community structure, nothing more. Whether that change is anyhow related to obesity you cannot tell. That is absolutely not telling you anything related to weight loss or lipids. 

Additionally, in the figure caption there is a panel d-f that are not present.

*In line 259-260 you speak of shorgum black rice polysaccharide?

*The text between lines 250-270 can benefit from rephrasing. You probably have some nice results, but the way is writen makes it difficult to visualize. You need to make differential analysis between normal diet and the others diets and one the other hand between high fat and high fat + polysaccharide. That will give you differentially abundant taxa that you can describe later on. 

What you have there, data displayed in figures and taxa named in the text, there is no description whether that differences are between normal diet and high fat or as consequence of adding the polysaccharide. Describe the statistical comparisson made.

Plus, if you assume that normal diet is giving you a reference microbiota, ideally the inclusion of the polysaccharide would revert back to that microbiota (what you try to justify in PCoA). Therefore, rather than just put there a lot of names, focus on finding such interaction: bacteria that is significantly different between normal diet and high fat and that after adding the polysaccharide reverts back to levels similar to that of normal diet.  Additionally, if there are differences in "scientifically proven" beneficial bacteria or harmful bacteria you can also describe them.

That data would look much easier to undertand in figures where the x axis is the treatment and the y axis the abundance, with 5 dots linked by a line showing the "evolution" and whether after treatment that particular genus is similar to normal diet. 

*Next paragraph can benefit from similar fixings. 

*Metabolites: The same comments I made for PCoA are applicable for PCA here. There is no clear separation, maybe more in figure e, but definetively not on a). They are all clustered in the center. Plus, it does not seem like 50 points. 

There is no description as to how you make PLS. What are you using as Y? Is it the model binomial, multinomial? 

*"According to p value and Impact value, the metabolic pathways related to obesity were screened(Table S5)." How do you know the metabolites related to obesity?

*"The experimental results show that SRP regulates the composition and function of gut microbiota and then changes the differential metabolites in feces to improve obesity." So far, your results have only showed that adding a polysaccharide changes microbial community structure and the production of some metabolites. Whether those metabolites or those species have any influence on obesity has not been proven, at least with the data provided there

*Discussion: there is no data on biodiversity. How can you claim that has improven?

I do not understand the bit about Bacteroides and Firmicutes. For starters, you mean Bacteroidetes not Bacteroides, the latter is a genus. Secondly, apparently, you claim that bacteroidetes (I guess) is negativelly related to obesity, but then you say that you decrease it? This section is poorly writen and explained. 

The manuscript needs a rather in depth rewriting process. In general, results do not prove that gut microbiota or their metabolites have any effect on obesity, just that they change as consequence of adding a fiber. Fiber will reduce glucose absorption and therefore insuline release and therefore favor anabolic processes such as lipogenesis. WIth your data, you can only claim, if so, that the fiber can improve plasma biomarkers, microbial profile and metabolite profile making them "healthier" (if that is the case). However, no direct link to obesity has been showed. 

Author Response

Dear Editors and Reviewers:

Thank you for your letter and the reviewer’s comments concerning our manuscript entitled "Metagenomics and untargeted metabolomics analysis of the effect of sporisorium reilianum polysaccharide on improving obesity". Those comments are valuable and helpful for revising and improving our paper, as well as the important guiding significance to our study. We have studied the comments carefully and have made the correction which we hope to meet with approval. We had revised our manuscript very carefully. Two colleagues with abroad research experience also helped us improve English writing.

In the current version, we have addressed all the comments and revised the manuscript accordingly. The details of revisions are listed below point-by-point and the main corrections in the manuscript were marked in red.

Thanks for your time and consideration.

Yours Sincerely,

Yunlong Guo

Reviewer 1

The authors evaluate the effect of a fungus polysaccharide on obesity using rats model. For this purpose, they study serum biomarkers, gut microbiota community structure and untargeted metabolomics. The study design is appropiate, however I do have some comments:

  • We greatly appreciate your constructive criticisms and valuable comments, which were of great help in revising the manuscript.
  1. Statistical methods are missing some pieces such as PCoA, correlations, etc. All statistical analysis, including those performed on metabolites, should be under the proper section.
  • We have revised the statistical analysis related to the experiment. Meanwhile, we have made modifications and corrections accordingly in the revised manuscript.
  1. Pathway analysis is not described.
  • The pathways of metagenomics and untargeted metabonomics were described in the results and discussion section. Meanwhile, we have made modifications and corrections accordingly in the revised manuscript.
  1. There is very little description on the DNA analysis. I assume is 16S by the lenght of the reads but is not specified. Also, how did you assigned taxonomies, did you grouped them in ASVs, OTUs? Which primers did you used? All the information is needed should the experiment be replicated
  • We revised the whole metagenomics sequencing process. Meanwhile, we have made modifications and corrections accordingly in the revised manuscript.
  1. Reported results regarding PCoA is kind of misleading. First, there are too few points when there should be 50 (50 rats). Second, there is no closeness between high fat diet + polysaccharide and normal diet, they are far apart and much closer to just high fat diet. Regardless, you are making an interpretation that is totally subjective, and it is hard to tell whether there is any  tendency as you claim. PCoA is just an exploratory method and unless you perform any additional statistical test that provides p values, you cannot make such an interpretation. 
  • Based on the body weight, serum lipid levels, liver enzyme functions, liver and white adipose tissue sections of rats results of this experiment, it was shown that the HFD-SRPH group had better results than the other two groups. Because of limited experimental budget and comparative analysis through related literature, the NC, HFD and HFD-SRPH groups were selected for metagenomics sequencing of rat feces, and 4 samples were selected for each group.
  1. You say: "The above results showed that obesity caused changes in the gut microbiota of rat feces, and showed that SRP could effectively regulate the gut microbiota of rat feces to achieve weight loss and lipid reduction". That is not supported at all by PCoA. You are just showing that introducing a polysaccharided midly changes gut microbiota community structure, nothing more. Whether that change is anyhow related to obesity you cannot tell. That is absolutely not telling you anything related to weight loss or lipids. 
  • We have made modifications and corrections accordingly in the revised manuscript.

  1. Additionally, in the figure caption there is a panel d-f that are not present.
  • Due to our negligence, there was minor issue in the figure caption of manuscript. Meanwhile, we have made modifications and corrections accordingly in the revised manuscript.
  1. In line 259-260 you speak of shorgum black rice polysaccharide?
  • We have made modifications and corrections accordingly in the revised manuscript.
  1. The text between lines 250-270 can benefit from rephrasing. You probably have some nice results, but the way is writen makes it difficult to visualize. You need to make differential analysis between normal diet and the others diets and one the other hand between high fat and high fat + polysaccharide. That will give you differentially abundant taxa that you can describe later on. 
  • We have made modifications and corrections accordingly in the revised manuscript.
  1. What you have there, data displayed in figures and taxa named in the text, there is no description whether that differences are between normal diet and high fat or as consequence of adding the polysaccharide. Describe the statistical comparisson made.
  • We have made modifications and corrections accordingly in the revised manuscript.

  1. Plus, if you assume that normal diet is giving you a reference microbiota, ideally the inclusion of the polysaccharide would revert back to that microbiota (what you try to justify in PCoA). Therefore, rather than just put there a lot of names, focus on finding such interaction: bacteria that is significantly different between normal diet and high fat and that after adding the polysaccharide reverts back to levels similar to that of normal diet.  Additionally, if there are differences in "scientifically proven" beneficial bacteria or harmful bacteria you can also describe them.
  • We have made modifications and corrections accordingly in the revised manuscript.
  1. That data would look much easier to undertand in figures where the x axis is the treatment and the y axis the abundance, with 5 dots linked by a line showing the "evolution" and whether after treatment that particular genus is similar to normal diet. 
  • The x-axis is different grouping, and the y-axis is relative abundance. The column plot of experimental results is more intuitive.
  1. Next paragraph can benefit from similar fixings. 
  • The x-axis is different grouping, and the y-axis is relative abundance. The column plot of experimental results is more intuitive.

  1. Metabolites: The same comments I made for PCoA are applicable for PCA here. There is no clear separation, maybe more in figure e, but definetively not on a). They are all clustered in the center. Plus, it does not seem like 50 points. 
  • Based on the body weight, serum lipid levels, liver enzyme functions, liver and white adipose tissue sections of rats results of this experiment, it was shown that the HFD-SRPH group had better results than the other two groups. In order to correspond to the groups of metagenome sequencing data, the NC, HFD and HFD-SRPH groups were selected for untargeted metabonomics analysis based on LC-HRMS of rat feces, and 10 samples were selected for each group. Figure 5a was the PCA in positive ion mode, yellow dots represent quality control samples, and relative aggregation indicates the stability of the experiment.
  1. There is no description as to how you make PLS. What are you using as Y? Is it the model binomial, multinomial? 
  • In this experiment, PLS-DA (partial least squares discriminant analysis) was used for multivariate statistical analysis. NC, HFD and HFD-SRPH groups have obvious separation in positive and negative ion mode. The displacement test plot in positive and negative ion modes shows that positive ion modes are R2Y=0.987 and Q2=0.935, and negative ion modes are R2Y=0.989 and Q2=0.945. R2Y reflects the stability of the model, and the closer the value is to 1, the better the stability of the model. Q2 reflects the reliability of the model. The closer the numerical value is to 1, the better the reliability of the model.
  1. "According to p value and Impact value, the metabolic pathways related to obesity were screened(Table S5)." How do you know the metabolites related to obesity?
  • Based on the body weight, serum lipid levels, liver enzyme functions, liver and white adipose tissue sections of rats results of this experiment, we speculated that the metabolites are related to obesity. Meanwhile, we have made modifications and corrections accordingly in the revised manuscript.
  1. "The experimental results show that SRP regulates the composition and function of gut microbiota and then changes the differential metabolites in feces to improve obesity." So far, your results have only showed that adding a polysaccharide changes microbial community structure and the production of some metabolites. Whether those metabolites or those species have any influence on obesity has not been proven, at least with the data provided there.
  • Due to our negligence, you misunderstood the experimental results. Meanwhile, we have made modifications and corrections accordingly in the revised manuscript.
  1. Discussion: there is no data on biodiversity. How can you claim that has improven?I do not understand the bit about Bacteroides and Firmicutes. For starters, you mean Bacteroidetes not Bacteroides, the latter is a genus. Secondly, apparently, you claim that bacteroidetes (I guess) is negativelly related to obesity, but then you say that you decrease it? This section is poorly writen and explained. 
  • Due to our negligence, you misunderstood the experimental results. Meanwhile, we have made modifications and corrections accordingly in the revised manuscript.
  1. The manuscript needs a rather in depth rewriting process. In general, results do not prove that gut microbiota or their metabolites have any effect on obesity, just that they change as consequence of adding a fiber. Fiber will reduce glucose absorption and therefore insuline release and therefore favor anabolic processes such as lipogenesis. With your data, you can only claim, if so, that the fiber can improve plasma biomarkers, microbial profile and metabolite profile making them "healthier" (if that is the case). However, no direct link to obesity has been showed. 
  • Firstly, we established a rat model of obesity induced by high-fat diet, after the intervention of SRP, the body weight, serum lipid levels, liver enzyme functions, liver and white adipose tissue sections of rats results of experiment showed that SRP could improve obesity. Secondly, We made metagenomics and untargeted metabonomics analysis of rat feces, and the results showed that the identified intestinal flora and metabolites may be related to obesity. We didn't express it clearly in the manuscript, which led to misunderstanding. Meanwhile, we have made modifications and corrections accordingly in the revised manuscript.

Reviewer 2 Report

The study presents high quality and deals with the important clinical issue, such type of study is needed.  I have only few small remarks that the authors should address properly.

I recommend the manuscript minor revision.

There are only some points to correct:

 - In the “objectives” paragraph, the aim is not clearly specified, although it is understandable when reading the whole article. Could You add one clear sentence about the intention, a problem that the article is trying to solve? Maybe a hypothesis, which will be confirmed or not in the conclusion section?

 - please provide the list of abbreviations

 - please provide the number of ethical approval

  • - introduction and discussion section need improvement; please provide information on how your results will translate into clinical practice; 

- in the discussion section please provide study's strong points  and study limitation section

- please correct typos

All the abovementioned issues are crucial for the credibility of the results. All the issues in the paper should  be solved, and another subsequent review.

I recommend the manuscript minor revision.

Author Response

Dear Editors and Reviewers:

Thank you for your letter and the reviewer’s comments concerning our manuscript entitled "Metagenomics and untargeted metabolomics analysis of the effect of sporisorium reilianum polysaccharide on improving obesity". Those comments are valuable and helpful for revising and improving our paper, as well as the important guiding significance to our study. We have studied the comments carefully and have made the correction which we hope to meet with approval. We had revised our manuscript very carefully. Two colleagues with abroad research experience also helped us improve English writing.

In the current version, we have addressed all the comments and revised the manuscript accordingly. The details of revisions are listed below point-by-point and the main corrections in the manuscript were marked in red.

Thanks for your time and consideration.

Yours Sincerely,

Yunlong Guo

Reviewer 2

The study presents high quality and deals with the important clinical issue, such type of study is needed. I have only few small remarks that the authors should address properly.

  • We greatly appreciate your constructive criticisms and valuable comments, which were of great help in revising the manuscript.
  1. In the “objectives” paragraph, the aim is not clearly specified, although it is understandable when reading the whole article. Could You add one clear sentence about the intention, a problem that the article is trying to solve? Maybe a hypothesis, which will be confirmed or not in the conclusion section?
  • In the abstract and conclusion, we added the suggestions of the reviewers. Meanwhile, we have made modifications and corrections accordingly in the revised manuscript.
  1. Please provide the list of abbreviations.
  • Abbreviations: ALT, Alanine aminotransferase; AST, Aspartate aminotransferase; HDL-c, High density lipoprotein cholesterol; HFD, High fat diet; HMDB, Human metabolome database; KEGG, Kyoto encyclopedia of genes and genomes; LC-HRMS, Liquid chromatography-high resolution mass spectrometry; LDL-c, Low density lipoprotein cholesterol; NCBI, National Center for Biotechnology Information; NR, Non-Redundant Protein Sequence Database; PCA, Principal component analysis; PCoA, principal co-ordinates analysis; PLS-DA,Partial least squares discriminant analysis; QC,Quality control; SRP,Sporisorium reilianum polysaccharide. Meanwhile, we have made modifications and corrections accordingly in the revised manuscript.
  1. Please provide the number of ethical approval.
  • All animal experiments were approved by the laboratory animal ethics committee of Changchun University of Chinese Medicine (Approval No.2021553) and performed in accordance with the regulations and guidance of this committeeMeanwhile, we have made modifications and corrections accordingly in the revised manuscript.
  1. Introduction and discussion section need improvement; please provide information on how your results will translate into clinical practice. 
  • In the abstract and conclusion, we added the suggestions of the reviewers. Meanwhile, we have made modifications and corrections accordingly in the revised manuscript.
  1. In the discussion section please provide study's strong points and study limitation section.
  • In the above, SRP has a significant effect on gut microbiota and metabolites in rats fed with high-fat diet, which can effectively increase the abundance of some beneficial bacteria or beneficial metabolites. However, there are still some limitations to be further studied in the regulation of host metabolism by SRP through gut microbiota. Whether the changes of small molecular metabolites in the gut microbiota and feces of the host after SRP intervention are caused by the changes of gut microbiota may need sterile animal models or fecal transplantation to verify. In the future, in-depth study of these aspects may provide a new perspective for functional food development and clinical research. Meanwhile, we have made modifications and corrections accordingly in the revised manuscript.

  1. Please correct typos.
  • We have corrected typos and corrections accordingly in the revised manuscript.
  1. All the above mentioned issues are crucial for the credibility of the results. All the issues in the paper should be solved, and another subsequent review.
  • We have corrected the above mentioned issues one by one and corrections accordingly in the revised manuscript.

Round 2

Reviewer 1 Report

The authors have answered my concerns.